# Nutritional Status in Spanish Adults with Celiac Disease Following a Long-Term Gluten-Free Diet Is Similar to Non-Celiac

**DOI:** 10.3390/nu13051626

**Published:** 2021-05-12

**Authors:** Catalina Ballestero-Fernández, Gregorio Varela-Moreiras, Natalia Úbeda, Elena Alonso-Aperte

**Affiliations:** 1Department of Pharmaceutical and Health Sciences, Facultad de Farmacia, Universidad San Pablo-CEU, CEU Universities, 28925 Alcorcón, Spain; cat.ballestero@ceindo.ceu.es (C.B.-F.); gvarela@ceu.es (G.V.-M.); 2Spanish Nutrition Foundation (FEN), C/General Álvarez de Castro 20, 1 pta, 28010 Madrid, Spain

**Keywords:** celiac disease, gluten-free diet, nutritional assessment, adults, dietary intake, nutrient intake, anthropometric measures, bone mineral density, physical activity

## Abstract

The only available treatment for celiac disease is life-long gluten exclusion. We conducted a cross-sectional age- and gender-matched study in 64 celiac adults on a long-term (>1 year) gluten-free diet and 74 non-celiac volunteers from Spain, using dietary, anthropometric, and biochemical parameters, as well as assessing bone mineral density and physical activity. Celiac adults had deficient intake (below 2/3 of the recommended intake) for folates, vitamin E, and iodine and low intake of calcium (below 80% of the recommended intake). Iron intake was also below 2/3 of the recommended intake in celiac women. Vitamin D intake was extremely low, and 34% of celiac patients had moderately deficient plasma levels. According to bone mineral density, celiac women may be more prone to osteopenia and osteoporosis. However, we found a perfectly analogous nutritional status scenario in celiac as compared to healthy volunteers, with the dietary deviations found being similar to those of the Spanish population, i.e., both groups followed a high-lipid, high-protein, and low-carbohydrate diet. Values for biochemical parameters were found within the reference ranges. Celiac disease had no influence on body weight, but body fat in celiac patients tended to be higher. According to our results, vitamin D, calcium, folates, vitamin E, iodine, and iron nutritional status should be specifically assessed and monitored in the celiac population.

## 1. Introduction

Celiac disease (CD) is an autoimmune systemic disorder triggered by the intake of gluten and its prolamins, which affects genetically susceptible individuals, causing progressive atrophy of the intestinal villi [1]. The only existing treatment is to follow a strict gluten-free diet (GFD) that repairs intestinal damage and restores adequate nutrient absorption [2]. The GFD should meet the recommended nutritional goals for energy and nutrients, just as it is for the general population.

The prevalence of CD in European and American populations of European descent is 1% [3]. In addition, it is estimated that there is a high percentage of undiagnosed cases [4]. So far, there are few studies that assess the overall nutritional status of people with celiac disease following a long-term GFD diet, but most of them agree that energy and nutrient intake are far from recommendations. Several studies agree on a higher intake of fats, proteins, and simple carbohydrates [5,6,7,8,9,10,11,12]. At the same time, some studies describe insufficient intake of fiber and carbohydrates, as well as of certain vitamins and minerals, possibly due to the exclusion of grains naturally rich in fiber, and the incorporation of commercial gluten-free products, whose content in refined flours and fats is higher than in their gluten-containing counterparts [13,14]. The high fat intake described in celiac population, together with a higher content of saturated fats and hydrogenated fatty acids in gluten-free products and a higher prevalence of overweight and obesity found in celiac population, makes it necessary to manage GFDs nutritionally to prevent health complications, such as the risk of developing metabolic syndrome and cardiovascular disease [15,16,17]. Interestingly, a higher glycemic index has been described in gluten-free products compared to their gluten-containing counterparts, since they are made with highly refined cereals and the amount of fiber is lower [13,18,19].

Furthermore, insufficient vitamin D and calcium intake are also documented in celiac population. This could be explained by the restriction of dairy products, due to a high prevalence of lactose intolerance in people with CD. In these cases, calcium and vitamin D supplementation, as well as using lactose-free products should be considered [2]. Low intake of calcium and vitamin D, together with the fact that low bone mineral density (BMD) is widely documented in CD patients [20,21,22], are further reasons why the contribution of diet to the nutritional status of people with CD should be assessed, to avoid the development of long-term bone alterations.

The reason why CD has been associated with low BMD is mainly due to the characteristic malabsorption, with the consequent deficiency in vitamin D and intestinal absorption of calcium. Other deficiencies in fat-soluble vitamins (A, K, and E) and minerals would also affect normal bone metabolism [23]. On the other hand, given the close hormonal interrelationship, calcium and vitamin D deficiencies stimulate the secretion of parathormone (PTH), and hyperparathyroidism would itself be another factor involved, since elevated levels of PTH have been associated with loss of bone mass by activation of bone resorption. Elevated PTH serum levels have been found in patients on a GFD [22].

Nonetheless, the available data on the effect of GFD on BMD provide discordant results, probably due to the different characteristics of the population studied in terms of age and sex, and due to the different amount of time on a GFD. Once the elimination of gluten occurs, the greatest gain in bone mass is established in the first year [24], and the GFD leads to a 5% increase in bone mass after one year of implementation [25]. However, in several studies [26,27], improvement in BMD after undertaking a GFD was only observed in patients who initially had secondary hyperparathyroidism, low serum calcium and vitamin D. Therefore, it may not be easy to normalize BMD in adults with CD [28]. In fact, there is a current debate on whether the GFD is sufficient to normalize bone alterations, or whether calcium and vitamin D supplementation should be recommended for these patients. Calcium fortification of gluten-free products has been suggested as a strategy to improve the calcium content in the diet of celiac patients [29]. Taken together, nutritional status may play a very relevant role in the maintenance and recovery of BMD in patients with CD in the long term, making it necessary to monitor intake of nutrients of vital importance for bone metabolism. Moreover, the same may be applied to physical activity, since people who are regularly active have greater bone mass and a lower number of fractures than those who lead a sedentary life.

The data available in the literature regarding body composition of people with CD following a GFD are not very clarifying of the real situation of this population group. Firstly, due to the variability in the age range of the populations studied, and secondly, due to the difference in the moment at which anthropometric and BMD measurements are made with respect to the beginning of the GFD. Nevertheless, there is some consensus on the favorable response of body composition to a GFD, normalizing situations of thinness, overweight, obesity and low BMD after the recovery of nutrient absorption functionality [30,31,32]. Classically, it has been described that patients with CD who follow a GFD, with a complete remission of the disease clinic, have lower BMI and body fat as compared to non-celiac [6]. This is also described in a recent review [33], which concluded that BMI is lower in subjects who have not yet begun the GFD compared to those who are on a GFD, and in celiac following a GFD compared to the population without the pathology. In Spain, a study carried out in the Basque Country has found different eating patterns between men and women, and a difference in anthropometric measurements. Thus, celiac women had lower BMI compared to the general population, low prevalence of overweight (6.5%) and no cases of obesity [9]. In the case of men, they found a higher prevalence of overweight (26.2%) and obesity (11.9%), as compared to celiac women, but still this prevalence was lower than in general population [34]. It should be noted that, in the adult population, the histological recovery of the intestinal mucosa is not as effective as in children.

On these premises, the aim of this study was to perform a thorough analysis of nutritional status in a Spanish CD adult population, including dietary quality, level of exercise, together with the analysis of body composition and biochemical parameters, all of which are necessary to describe how long-term adherence to a GFD impacts on nutritional status.

## 2. Materials and Methods

### 2.1. Subjects

This is a cross-sectional age- and gender-matched study in celiac and non-celiac adult volunteers. People with CD were located through a patient’s association (Asociación de Celíacos y Sensibles al Gluten) in Madrid (Spain). The criteria for inclusion in the study for celiac were confirmed diagnosis of celiac condition, adherence to a gluten-free diet for more than one year, absence of associated diseases, and not taking nutritional supplements. In the non-celiac group, the inclusion criteria were not being diagnosed with any chronic disease, not having symptoms or signs of digestive disease on a regular basis, and not taking nutritional supplements. All participants had to be between 18 and 59 years old. In both cases, volunteers who did not meet the inclusion criteria were excluded, as well as women who were pregnant or breastfeeding. In the case of the non-celiac, those volunteers who tested positive in the anti-tissue transglutaminase IgA class antibodies (AAtTG) antibody test were also excluded.

Taking into account previous studies in Spain and considering a confidence interval of 95%, an error α of 5%, an error β of 20%, a power of 80%, and a case/control ratio of 1:1, using the EpiInfo v.7 software, a total sample of 110 adults was calculated. Predicting a loss rate of 20%, an initial sample of 75 cases and 75 controls aged 19–59 years was proposed. The final sample included 138 volunteers; 64 had celiac disease and 74 were healthy volunteers. The percentage of lost subjects due to not adequately meeting the participation requirements was 8%. All recruited volunteers completed the study.

The protocol was approved by the Ethics Committee for Human Studies at Universidad San Pablo-CEU (Authorization number 124/16/09). The project was conducted in accordance with legal requirements and guidelines for good clinical practice, as well as the World Medical Association Declaration of Helsinki on Ethical Principles for Medical Research Involving Human Subjects (revised in October 2008). All volunteers (celiac and non-celiac) were informed and provided their written consent to participate in the study. Anonymity and personal data protection were guaranteed, as established by the Spanish Organic Law on the Protection of Personal Data (3/2018 of December 5th).

### 2.2. Analysis of Dietary Intake and Eating Habits

A dietitian and trained anthropometrist interviewed the participants. Three 24-hour dietary records were analyzed. The first record was collected during the first face-to-face analysis session, and the two remaining ones were collected by telephone with a one-month interval between each survey. One of them was always carried out on a holiday, following the methodological recommendations of the EFSA (European Food Safety Authority) E-Menu methodology [35]. The interview was conducted with the help of home measurements of food servings, and photographic books to estimate quantities, thus making it easier for participants to describe their dietary intake.

The dietary records were analyzed, by means of the computer software DIAL^®^, to transform food intake into energy and nutrient consumption. In the case of gluten-free products, nutrient composition was taken from a gluten-free food composition database previously developed in our research group [36]. The results were compared with the Recommended Intake of Energy and Nutrients for the Spanish Population [37], and with the Nutritional Objectives established by the Spanish Society of Community Nutrition (Sociedad Española de Nutrición Comunitaria, SENC) [38]. Adherence to recommendations was calculated using the following formula: (actual intake/recommended intake) ×100.

For the analysis of eating habits, food consumption frequency questionnaires based on validated questionnaires [39] were applied. The questionnaires provided information on the number of meals per day and the frequency of food consumption by groups (vegetables, fruits, dairy products, cereals, cookies and pastas, nuts., etc.) Subsequently, frequency data were compared with daily recommended servings from each food group, according to the Spanish Society of Community Nutrition (SENC).

### 2.3. Anthropometric Measures

All anthropometric measurements were taken according to the International Standards for Anthropometric Assessment, issued by the International Society for the Advancement of Kinanthropometry (ISAK) [40]. Measures were taken by a level 1 anthropometrist, on the right side of the volunteers, and a minimum of two times. A third measurement was taken when both measurements differed by more than 5%. The room where the measurements were taken had sufficient privacy and temperature was set to be comfortable for the volunteers. The materials used were the following: SECA^®^ portable stadiometer, accuracy 1 mm; anthropometric box; SECA^®^ digital homologated scale, accuracy 100 g; CESCORF metallic anthropometric tape, accuracy 1 mm; Harpenden^®^ plicometer, accuracy 1 mm; Cescorf^®^ small bone caliper. The following anthropometric measurements were taken according to ISAK methodology, comprising the Complete Restricted Profile [40]: height, weight, subcutaneous folds (triceps, biceps subscapular, iliac crest, supraspinal, abdominal, medial calf, and frontal thigh); girths (relaxed arm, contracted arm, waist, hip, calf); bone breadths (biepicondyle diameter of the humerus and biepicondyle diameter of the femur).

Using these parameters, the following indexes were calculated: Body Mass Index (BMI) (weight (kg)/height^2^ (m)) and body fat percentage. Body fat was calculated from the sum of the measured triceps, biceps, subscapular and supraspinal folds, using the Durning and Womersley formula [41].

### 2.4. Bone Mineral Density

The analysis of bone mineral density (BMD) was performed by a Hologic ultrasonic densitometer model Sahara^®^, taking the measurement on the calcaneus. The densitometer was calibrated before each measurement according to the manufacturer’s instructions. This technique is very simple and non-invasive since it is done by placing a bare foot in the densitometer. The ultrasound measures bone mechanical properties, such as attenuation or BUA (Broadband ultrasound attenuation), and the speed with which the sound passes through the bone or SOS (Speed of Sound), providing information on the elasticity and density of the bone [42]. In addition to these parameters, for the diagnosis of osteoporosis, the T-Score, which takes as a reference the maximum average BMD reached at around 30 years of age, has been evaluated. Reference values consider normality when the BMD is greater than −1 standard deviation from peak bone mass (T score ≥ −1 SD); osteopenia when the BMD is between −1 and −2.5 standard deviations from peak bone mass (T score ≤ −1 and ≥ −2.5 SD), and osteoporosis when the BMD is less than −2.5 standard deviations from peak bone mass (T score ≤ −2.5 SD).

### 2.5. Blood Parameters

The analytical determinations were carried out by authorized personnel of a certified clinical laboratory in Madrid, Spain (*Megalab S.L.*), extracting a sample of fasting venous blood. The following parameters were determined:

Hematological: Red blood cell count, hemoglobin, hematocrit, mean corpuscular volume (MVC), mean corpuscular hemoglobin (MCH), mean corpuscular hemoglobin concentration (MCHC), red cell distribution width (RDW), platelet count, median platelets volume (MPV), leukocyte differential count (lymphocytes, monocytes, neutrophils, eosinophils, basophils).

General biochemistry: iron, basal glucose, homocysteine, total calcium, phosphorus, cholesterol, triglycerides, HDL-cholesterol, LDL-cholesterol.

Vitamin metabolism: folate, 25-OH Vitamin D.

Malabsorption markers: albumin.

Hormonal alterations: parathormone.

### 2.6. Physical Activity

For the evaluation of physical activity, the internationally validated IPAQ (International Physical Activity Questionnaire) was used [43]. It includes a sequence of questions about the activities carried out in the last seven days, positively evaluating the intensity and time spent in each type of activity. As described in the methodology of analysis of the questionnaire, according to the intensity of the activity carried out (walking, moderate activity or vigorous activity), the time employed is multiplied by a factor (3.3; 4 or 8 respectively), and by the number of days in which the activity has been carried out, and in this way, we obtain a final result in Metabolic Equivalents of Task (METs), a unit used to estimate the metabolic expenditure for a specific activity. This calculation allowed us to categorize the performed physical activity as insufficient, moderate, or high.

### 2.7. Data Collection and Statistics

A protocolized collection of data was conducted using templates designed for this purpose.

The statistical analysis was carried out using the SPSS version 24 program. Normal distribution in quantitative variables was checked by means of the Kolmogorov–Smirnov test. To unify data, results are expressed as median (p25–p75). The comparison between the two groups (celiac and non-celiac), both in the global sample, and divided by gender and age groups, was carried out by means of the Mann–Whitney U test and the Student’s t-test, as appropriate in the sample, establishing *p* ≤ 0.05 as the significance level.

Categorical or qualitative variables (frequencies) were compared through Pearson’s chi-squared test, using contingency tables. A significant value of *p* ≤ 0.05 was considered. For the analysis of correlations between variables with normal distribution, Pearson’s correlation was used, otherwise, Spearman’s correlation was used.

## 3. Results

As shown in Table 1, 67.2% of celiacs were women, and 32.8% were men. In the non-celiac group, 66.2% were women and 33.8% men. The average age in the group of women was 39.17 ± 10.62, and in the case of men, 38.58 ± 9.61.

### 3.1. Dietary Habits and Nutrient Intake

According to general intake habit/food behavior, subjects in this study consumed four meals a day on average. Most of the celiac volunteers (93.8%) declared to follow a strict gluten-free diet. The results of the food frequency questionnaire are summarized in Table 2. In the group of celiacs, daily consumption of vegetables was higher, as well as weekly consumption of legumes as compared to the non-celiac group. All celiac subjects reported drinking at least one serving of vegetable drinks per week, compared to 76.6% of non-celiac. The same was observed for pickles, which were eaten at least once a week by 69% of the sample of celiac compared to 40.6% of non-celiac (*p* = 0.001).

However, the consumption of cereals and derivatives is significantly higher in the non-celiac group, as can be expected considering the limitations of the gluten-free diet for this food group.

When considering gender, celiac men had a significantly higher consumption of vegetables and legumes, as well as a higher consumption of meat and eggs, as compared to non-celiac. In the group of women with CD, the consumption of cereals and derivatives was significantly lower.

When comparing with the daily/weekly recommendations of food group servings issued by the Spanish Society of Community Nutrition (SENC) [44], food consumption frequencies in both celiac and non-celiac did not meet recommendations. Number of servings of dairy per day did not reach the three recommended servings, and the same happens with intake of fruits, vegetables, cereals, and legumes, which are insufficient, except in the group of celiac men, who met the legumes weekly intake recommendations. On the contrary, consumption of meat, eggs, fish, and seafood exceeded the recommendations in celiac men, but this was not the case in the rest of the volunteer groups. In the case of pastries and cakes, consumption is higher than recommended, both in women and men, celiac and non-celiac.

No significant differences were found between the number of volunteers who declared they had been breastfed (celiac: 76.6%; non-celiac: 78.4%) and those who had not (celiac: 12.5%; non-celiac: 14.9%), nor in the time of breast-feeding they declared to have received, although the average time was higher in the non-celiac group (5.29 ± 6.86 months) than in the celiac group (3.79 ± 2.60 months).

Recommended macronutrient distribution for the Spanish population is 50–55% of total energy from carbohydrates, 10–15% from proteins and up to 35% from lipids [38]. In the present study (Table 3), percentage contribution of carbohydrates to total energy in celiac was similar to the non-celiac group, and both were below the recommended values. Percentage contribution of proteins to total energy was also similar in all groups but exceeded recommendations. Finally, the contribution of lipids to total energy was higher compared to recommendations in all the groups studied. A slightly lower contribution of lipids was observed in the group of men with CD with respect to the non-celiac group, thus, lipid consumption in celiac men was closer to recommendations. PUFA intake was significantly lower in the celiac group compared to the non-celiac. As for the contribution of simple sugars (mono- and disaccharides) to total energy, no significant differences were found between groups; however, a higher consumption was observed in the groups with CD, both men and women, in both of which the recommendations (<6% of the total energy) were surpassed.

Table 3 also shows the contribution of different lipids to total energy intake, intake of cholesterol and different types of fat. A high contribution of SFA was observed, as compared to recommendations (7–8% of total energy from SFA), both in the groups with CD and in the non-celiac group. In the case of men, percentage contribution of SFA to total energy in celiac was significantly lower as compared to non-celiac, although still above recommendations. In relation to PUFA and MUFA, nutritional objectives for the Spanish population were met more closely (5% and 20% of total energy from PUFA and MUFA, respectively). MUFA percentage contribution to total energy was lower in celiac, but not statistically significant when compared to non-celiac. There are no significant differences in MUFA, cholesterol, trans fatty acids, ω6-fatty acids, ω3-fatty acids, EPA, and DHA intake between the groups with CD and the non-celiac groups, both when comparing the total sample and when dividing it by gender. Cholesterol intake is above the recommendations (<300 mg/day) in all study groups, except for women in the non-celiac group.

In the same way, no significant differences were found between volunteers with CD and non-celiacs in fiber intake. The nutritional objectives for the Spanish population propose a minimum intake of 22 to 25 g/day of fiber in women and a minimum of 25 to 35 g/day for men [38]. Celiac men and women have a higher fiber intake and are, therefore, closer to meeting the recommendations as compared to non-celiac.

Table 4 and Table 5 show the adequacy of nutrient intake to reference intake values for the Spanish population, calculated according to the formula %RI = actual intake/recommended intake × 100. It is relevant that, in both celiac and non-celiac, actual intake does not cover 2/3 of the recommended intake of energy, folates, vitamin D, vitamin E, calcium, iodine, zinc, and magnesium. In the case of iron, intake is also low in the total sample and more specifically in the group of women, both celiac and non-celiac. The data concerning vitamin D intake are especially worrying, since their intake does not cover 30% of the recommended intake in any of the study groups. When comparing intake between celiac and non-celiac populations, intake of vitamins C and A in celiac are significantly higher and reach a higher degree of compliance with recommendations. Likewise, vitamin K intake was significantly different, being higher in the group of celiac, especially men. In the men’s group, intake of folate and pyridoxine is higher in the celiac population compared to non-celiac; furthermore, folates intake in the non-celiac group fails to reach recommendations, whilst intake is above 2/3 of the recommended intake in celiac men. On the other hand, intake of phosphorus in celiac is significantly lower than in the non-celiac group, although the recommended intake was accomplished in all groups.

### 3.2. Anthropometric Measurements

Celiac men presented lower weight, lower body fat, and lower BMI when compared to their non-celiac counterparts. In the women’s group, significant differences are only observed when comparing body fat, which was higher in celiac women (Table 6). There were no significant differences in height. Figure 1 shows the classification of weight status according to body mass index (BMI). In the group of celiac men, a significantly higher proportion of subjects fell in the BMI range classified as normal weight, as well as in the underweight group, as compared to non-celiac. In the non-celiac group, a significantly higher proportion of subjects was found in the ranges of overweight grade I, overweight grade II, and obesity. For women, the proportion of celiac women in the normal-weight range was higher than in the non-celiac group, whilst a higher proportion of non-celiac women were classified in the underweight category (Figure 1).

Because of the influence of age in body fat composition, data on body fat were categorized in two different age groups, young adults (19 to 39) and old adults (40 to 59), and was used to categorize body weight as insufficient, adequate, overweight, or obesity [45]. Data show that in the youngest age group (10–39 years old), women with CD had a higher prevalence of overweight (25%) and obesity (20%) than non-celiac (4.3% and 0%, respectively), and the prevalence of underweight was higher in the non-celiac group (26.1% vs. 0%). Within the healthy category, we found 55% of celiac women versus 69.6% of non-celiac women. In men, it was the non-celiac group who presented the highest prevalence of overweight and obesity. In the 40–59 years old group, the difference in men with CD with respect to the non-celiac group was relevant and significant, since celiac presented a lower prevalence of overweight (15.4% vs. 33.3%) and obesity (23.1% vs. 53.3%) and a higher percentage of subjects in the healthy category (61.5% vs. 26.7%).

### 3.3. Bone Mineral Density

Bone mineral density (Table 7) did not differ between celiac and non-celiac, neither in the total sample group nor divided by gender.

The following Figure 2 shows bone mineral density status according to T-score. In men, 29% of celiac and 32% of non-celiac presented values suggestive of osteopenia, and prevalence was similar in both groups. In the case of women, the data obtained showed less favorable results for the group of women with CD, since this group presented a greater and significant risk of osteopenia (44.9%) and osteoporosis (2%) compared to the group of non-celiacs. No relationship was found between these results and blood or intake data for critical nutrients for bone health (calcium, vitamin D, phosphorus) or physical activity.

### 3.4. Blood Parameters

The data referring to the study of the hemogram and white cell series (Table 8) show that all parameters studied (red blood cell counts, hemoglobin, hematocrit, MCV, MCH, MCHC, RDW, and MPV, leukocyte differential count (leucocytes, lymphocytes, monocytes, neutrophils, eosinophils, and basophils)) were within the reference ranges for the population studied. The differences between celiac and control show a significantly lower value in the number of red blood cells in celiac men, and higher values in MCV and MCH. No significant differences were found in the results for women. All the values for the white series were within the reference ranges. Numbers of leukocytes and neutrophils were significantly lower in celiac men as compared to non-celiac.

Table 9 shows the data referring to other biochemical parameters related to nutritional status, and all values were within the reference ranges. Significant differences were found for plasma triglyceride levels, being lower in the group with CD compared to the non-celiac group. On the other hand, a higher LDL/HDL ratio was observed in the group of women with CD compared to the non-celiac group. Finally, plasma albumin levels were significantly higher in the group of men with CD compared to the non-celiac group. In the rest of the blood parameters studied, no significant differences were found between celiac and non-celiac. Plasma levels of folate and homocysteine were similar in celiac and non-celiac, but the prevalence of hyperhomocysteinemia was higher, but not significantly, in the non-celiac group (8.1% vs. 1.6%).

Although we found no significant differences in the plasma values of 25-OH vitamin D, and the median of all the groups was within the reference ranges, we analyzed the distribution of the participants according to reference plasma levels (severe deficit, moderate deficit, recommended value, or excess). The results showed that plasma levels of this vitamin are found to be moderately deficient (10–30 ng/mL) in a greater proportion in the control group, both in the overall sample (41.9% in non-celiac and 34.4% in celiac), and when segregated by gender (men: 44% in non-celiac and 28.6% in celiac; women: 40.8% in non-celiac and 37.2% in celiac). Therefore, the high prevalence of moderate vitamin D deficiency in the general population is of concern, but according to our results, there is no greater deficiency in the group with CD as compared to the control group.

### 3.5. Physical Activity

The data obtained from the IPAQ assessing physical activity were used to estimate METs (min/week), and to categorize physical activity as insufficient, moderate, or vigorous according to the scoring protocol of the IPAQ [43]. In none of the cases were significant differences found when comparing physical activity performed by celiac and non-celiac, and most of the studied volunteers fell in the moderate to vigorous physical activity range (Figure 3). According to IPAQ [43], moderate physical activity equals three or more days of vigorous physical activity for at least 20 min per day, or five or more days of moderate physical activity and/or walking for at least 30 thirty minutes per day or five or more days of any combination of walking, moderate or vigorous physical activity achieving at least a total of 600 METs. Vigorous physical activity equals vigorous physical activity at least three days per week achieving a total of at least 1500 METs or seven days of any combination of walking, moderate physical activity, and/or vigorous physical activity, achieving a total of 3000 METs.

## 4. Discussion

As far as we know, this is the first study in which a complete evaluation of the nutritional status, through dietary and body composition analysis, as well as biochemical and physical activity measures, has been carried out in Spanish adults diagnosed with celiac disease, once a long-term gluten-free diet has been established at least for a year. In addition, data were compared with non-celiac healthy subjects, according to gender, to detect possible deficiencies and nutritional imbalances in the celiac population.

Patients who follow a GFD necessarily have to exclude many carbohydrate-containing foods with gluten, and it has been postulated that this restriction may lead subjects with CD to make inappropriate choices, and to prefer foods with a high caloric content and a higher proportion of fat and protein [2,16]. In addition, studies show that commercial gluten-free products are often of poorer nutritional quality than their gluten-containing equivalents [36,46,47]. In previous published studies, patients with CD consumed more fat (especially saturated), protein, and simple carbohydrates but less fiber and micronutrients, such as iron, calcium, and vitamin D, than recommended [5,7,16,48,49], as well as compared to healthy subjects [11,15,50]. As reviewed by Penagini et al. [18], the most common nutrient deficiencies found in celiac patients on a GFD included fiber, iron, folate, niacin, vitamin B12, and riboflavin.

In our study, no relevant differences were found in the contribution of macronutrients to total energy when compared to the non-celiac group, however, the same does not apply to vitamin and mineral intake. According to the macronutrient distribution profile, all groups showed high lipid, protein, and simple carbohydrates (mono- and disaccharides) intake, as well as low carbohydrate intake, as compared to the Spanish Nutritional Objectives [38]. Low-carbohydrate plus high-lipid and -protein diets are characteristic of the Spanish population, as shown by the recent national dietary surveys ANIBES and ENIDE [51,52]. According to our results, this same trend is observed in CD patients. Moreover, although celiacs need to avoid some carbohydrate-rich cereals, their carbohydrate intake is close to that of non-celiacs.

When analyzing the quality of lipid intake, none of the groups met recommendations, since the contribution of saturated fatty acids (SFA) to total energy intake was higher, and the contribution of mono-unsaturated fatty acids (MUFA) and poly-unsaturated fatty acids (PUFA) was lower, as compared to guidelines. The intake of PUFA was significantly lower in celiac patients compared to non-celiac subjects, as found in our previous study in children [53]. High intake of SFA and cholesterol (above 300 mg/day), as well as low intake of MUFA and PUFA, is also common in Spanish and European population [51]. PUFA intake is quite relevant since people with CD could benefit from their therapeutical anti-inflammatory effect [54,55,56].

In the case of fiber, we did find a slightly higher intake in people with CD, especially men, whose fiber intake was closer to meeting the recommendations. These data contrast with those found in previous reviews [13,18], which describe lower fiber intake in people with CD compared to healthy subjects. This is also the conclusion of a study carried out in Italy on 39 celiac adults (21–45 years of age) compared to a control group [10].

When assessing vitamin and mineral intake in celiac adults, we found severely deficient intake of vitamin D, deficient intake (below 2/3 recommended intake) of folates, vitamin E, and iodine, and low intake of calcium. Iron intake was also low (below 2/3 recommended intake) in celiac women. Nutrient intake in celiac patients, especially for vitamin D, folate, and iron, should be assessed, because some studies show a higher risk of bone disease [21,28,29,57,58] and anemia [59] in celiac population. Interestingly, in the present study we did not find lower intake of folate, vitamin D, or iron in celiac as compared to non-celiac. Furthermore, low intake of vitamin D, folate and iron is also commonly described for the general population [51]. When analyzing nutrient intake in celiac, some underestimation of vitamin and mineral intake should be accounted for, since data on gluten-free products’ composition are scarce, and almost non-existent for minerals and vitamins [36].

Folate intake in celiac men was significantly higher than that described in non-celiac, in contrast to what we found in our previous study in children and the young population [53], and what previous studies state [8,9,13,14,60]. The main sources of folate in the Spanish diet are vegetables, legumes, fruits, and milk and its derivatives [61], all of which do not need to be excluded in the GFD. Therefore, the higher vitamin intake may be attributable to different eating patterns. In fact, celiac men in our study consumed more vegetables and legumes, thus leading to higher intake of folate, as well as vitamins C, A, and K, which are also present in these food groups. This observation suggests that adults substitute gluten-containing foodstuffs with other food groups and not for commercial gluten-free products, as we found in the case of children and adolescents [53].

Taken all together, we did not find significant differences in nutrient intake in celiac as compared to non-celiac, with the dietary habits found being similar to those of the Spanish population. A similar previous cross-sectional study in Spain found different results, showing significantly different intake between celiac and healthy people for fat, protein, simple carbohydrates, iron, calcium, and vitamin D [11]. Another study in Spain analyzed diets in celiac women [9] as compared to reference values, not to a control group, showing similar deviations. Several reviews also describe nutritional deficiencies in the GFD [13,18].

Some authors attribute these differences to gluten-free product consumption [12]. To explain the difference to other studies, we propose that gluten-free diets in celiac population may have improved because of a closer clinical assessment and follow-up of patients, and availability of more varied gluten-free products. In this sense, the Spanish Ministry of Health, Social Services and Equality published a clinical guide for the early diagnosis of CD [62], focusing on the early detection of CD, but also addressing key issues such as treatment, clinical monitoring of patients, refractoriness, and malignancy, and follow-up and monitoring of CD patients on a gluten-free diet. Patients’ associations also contribute to nutritional education for early-diagnosed patients. Finally, we have also demonstrated a slight reformulation in fat composition and salt reduction in gluten-free products [36].

There is a consensus in the fact that a gluten-free diet recovers the intestinal villi and the absorption function in celiac patients. Thus, the values of blood parameters in people with CD who follow a long-term GFD should be similar to those of the general population. In the present study, values for biochemical parameters were found within the reference ranges, both in the celiac group and in the non-celiac group. Celiac men presented a smaller number of red blood cells, but higher mean corpuscular volume and hemoglobin, as compared to controls. Moreover, the number of leukocytes and neutrophils was also smaller in the group with CD, and especially in men. Nonetheless, since all values are within reference ranges, we conclude that there is no indication of abnormality in celiac patients on a long-term gluten-free diet, although this aspect should not be ruled out in some individuals. Conversely, some studies describe common alterations in people with CD, such as vitamin D deficiency [63,64] and anemia [59]. Other studies describe an altered lipid profile, due to diets with a high fat and protein content and a decreased consumption of complex carbohydrates [9,11]. In this context, some authors [13,18,19] describe a higher glycemic index for gluten-free products that could lead to different metabolic alterations. Our results could be again related to an improvement in the adherence to gluten-free diets and improved dietary habits.

Plasma homocysteine levels were within reference ranges, and no significant differences have been found between the prevalence of hyperhomocysteinaemia (>15 µmol/L) in celiac and non-celiac (1.6% in CD vs. 8.1% in non-celiac), although prevalence was higher in the non-celiac group. Again, these results are in contrast with those of another study published in 2002 that describes higher homocysteine levels in 30 individuals with CD compared to the general population [60].

Plasma values of 25-OH-vitamin D were not significantly different between celiac and non-celiac, but we found levels indicative of moderate vitamin D deficiency (values between 10–30 ng/mL) in 41.9% of the non-celiac, and in 34.4% of the celiac. In a more detailed analysis by gender, moderate deficit reaches a prevalence of 44% and 40,8% in healthy men and women, and 28.6% and 37.2% in celiac men and women. Again, our data agree with what is found in the general population, where according to the Kreutz study [65], vitamin D deficiency levels are found in up to 50% of healthy individuals. Vitamin D status proves to be slightly better in celiac than in non-celiac, and this could be due to a higher intake of fish products, a good source of vitamin D, in celiac men.

Studies on anthropometric parameters in people with CD after one year following a GFD provide diverse data. Valletta et al. [66] show an increase in the frequency of overweight, an effect that was also found by Mariani et al. [5] and Norsa et al. [67], who propose that this is explained by the normalization of the intestinal mucosa and the correct absorption of nutrients, accompanied by an unbalanced diet with increased consumption of fats and sugars from GFP. However, other studies show that a good compliance with the GFD had a positive effect on body composition [48,49,50] with a normalization of BMI, both in underweight and overweight subjects. Thus, Reilly et al. [68] and Bambrilla et al. [30] observed a decreased prevalence of overweight and obesity in patients following a GFD. According to the results of our study, we found a different situation in men and women with CD compared to the non-celiac groups. Thus, celiac men had lower body weight and fat, and lower BMI compared to non-celiac. On the contrary, celiac women presented a higher percentage of body fat and higher prevalence of overweight and obesity, especially the youngest, although a greater part of them could be classified as normal weight according to BMI. Therefore, we did not find a clear deleterious effect of CD and GFD on weight status, but special attention should be paid to the female gender according to our results.

In people with CD, the risk of suffering lower BMD is well-documented [20,58], mainly due to the characteristic malabsorption of calcium and vitamin D, which results in a decrease in bone mineral content. This means that in many cases, people with untreated CD during the critical periods in which maximum bone mineralization occurs, do not reach an adequate peak of bone mass, causing growth retardation in children and adolescents and increasing the risk of osteopenia and/or osteoporosis in adults [20,69]. In our study, we found no significant differences in BMD between men and women with CD and the non-celiac homologous group, but celiac women have a higher risk of osteopenia and osteoporosis according to the T-Score. Previous studies showed that adherence to the GFD is an important contributor to having a lower degree of bone anomalies [70], and can help to recover a normal BMD [71]. In our study, 93.8% of celiac claim to follow a strict gluten-free diet.

Finally, according to the scores obtained for physical activity, most volunteers engage in moderate to intense physical activity, which has probably contributed to a good nutritional status in our study group, since a physically active lifestyle can compensate for the insufficient intake of calcium and vitamin D found in all volunteers.

### Strengths and Limitations

The major strength of this study is that it carries out a complete nutritional assessment, including dietary intake and food habits, blood status, body composition, and physical activity. However, some limitations must be noted. Firstly, it is a cross-sectional study and, consequently, the present results must be interpreted with caution. Secondly, physical activity levels were self-reported (through the IPAQ) instead of objectively measured using accelerometry. Thirdly, there is an important limitation of data on micronutrient composition of commercial gluten-free products. Finally, this is not a randomized study and people participated voluntarily, meaning that bias as a result of more health-conscious volunteers must be considered. The low sample size must also be considered.

## 5. Conclusions

In conclusion, we found a perfectly equivalent nutritional status scenario in celiac adults as compared to healthy volunteers, with the dietary deviations found being similar to those of Spanish population, i.e., both groups followed a high-lipid, high-protein, and low-carbohydrate diet, ingested too many sugars and low amounts of vitamin D, folates, vitamin E, iodine, calcium, and iron. We did not find hematological signs of anemia, malnutrition, bone alterations, or any pathological situation attributable to CD, except that celiac women may be more prone to osteopenia and osteoporosis. Celiac disease had no clear influence on body weight, but women are also more likely to have more body fat. Nonetheless, vitamin D, calcium, folate, vitamin E, iodine, and iron nutritional status should be assessed and monitored in the celiac population.

## Figures and Tables

**Figure 1 nutrients-13-01626-f001:**
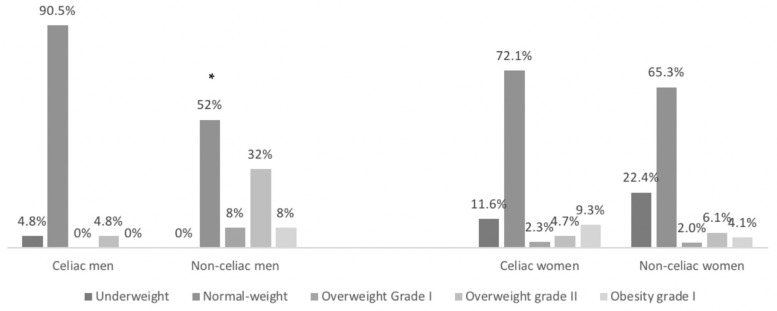
Classification of weight status according to BMI in celiac and non-celiac Spanish men and women. The results are shown as a percentage of subjects classified in each BMI category. Men: Pearson’s chi-squared, *p* = 0.023 *.

**Figure 2 nutrients-13-01626-f002:**
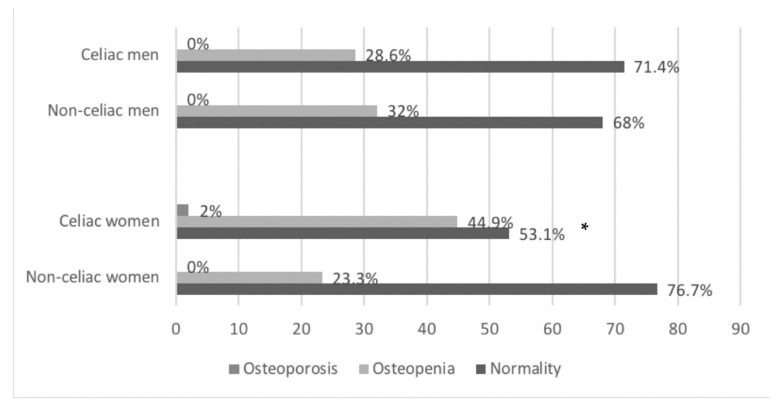
Bone mineral density status according to the T-score in celiac and non-celiac Spanish men and women. The results are shown as a percentage of subjects in each of the T classification categories. Men: Pearson’s chi-squared, *p* = 0.801. Women: Pearson’s chi-squared, *p* = 0.0500 *.

**Figure 3 nutrients-13-01626-f003:**
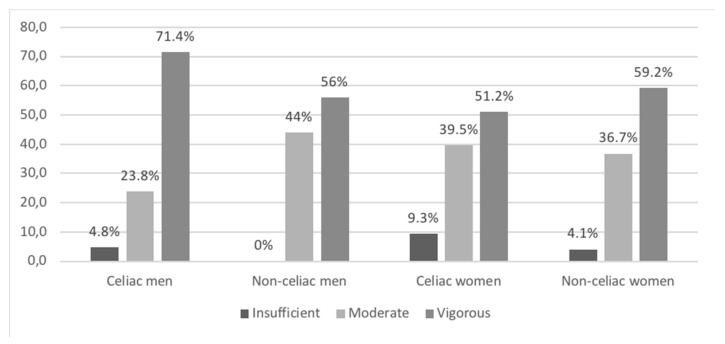
Level of physical activity performed by celiac and non-celiac Spanish men and women. The results are expressed as a percentage of participants classified in each category of physical activity. No significant differences were found between celiac and non-celiac. Men: Pearson’s chi-squared: 0.228; women: Pearson’s chi-squared: 0.530.

**Table 1 nutrients-13-01626-t001:** Description of the sample by gender.

	Celiac	Non-Celiac	Total	Age (Years)
Women	43 (67.2%)	49 (66.2%)	92 (66.7%)	39.17 ± 10.62
Men	21 (32.8%)	25 (33.8%)	46 (33.3%)	38.58 ± 9.61
Total	64	74	138	

Results are expressed in total number of participants and corresponding percentage in each case. Age is expressed as mean ± SD.

**Table 2 nutrients-13-01626-t002:** Frequency of food consumption in celiac and non-celiac Spanish men and women, expressed as number of servings per day/week.

	Total Sample			Men			Women			**SENC ^1^*Recommendations***
	Celiac (*n* = 64)	Non-Celiac (*n* = 74)	*p*	Celiac (*n* = 21)	Non-Celiac (*n* = 25)	*p*	Celiac (*n* = 43)	Non-Celiac (*n* = 49)	***p***	
Dairy (servings/day)	1.6 (0.9–2.2)	1.6 (1.0–2.3)	n.s.	1.6 (0.9–2.3)	1.7 (1.0–2.5)	n.s.	1.6 (0.9–2.3)	1.6 (0.9–2.3)	n.s.	2–3 servings/day
Fruits (servings/day)	1.7 (1.1–2.8)	1.7 (1.0–2.3)	n.s.	1.8 (1.2–3.2)	1.2 (0.8–2.3)	n.s.	1.7 (1.1–2.6)	1.7 (1.0–2.4)	n.s.	3–4 servings/day
Vegetables (servings/day)	1.1 * (1.1–1.5)	1.1 (0.8–1.5)	0.047	1.1 * (0.9–1.3)	0.8 (0.8–1.1)	0.044	1.3 (1.1–1.7)	1.1 (1.1–1.5)	n.s.	2–3 servings/day
Legumes (servings/week)	1.7 * (1.5–3.7)	1.5 (1.5–2.2)	0.009	2.00 * (1.5–4.0)	1.5 (1.5–1.7)	0.042	1.5 (1.5–3.0)	1.5 (1.5–3.0)	n.s.	2–4 servings/day
Meat and eggs (servings/week)	8.5 (4.0–11.0)	8.5 (4.0–11.0)	n.s.	11.0 * (8.2–12.5)	7.0 (6.0–11.0)	0.014	8.0 (6.0–9.5)	8.5 (6.7–11.0)	n.s.	White meats, 3 servings/week Eggs, 3 servings/week
Fish and seafood (servings/week)	4.0 (3.0–6.0)	3.2 (3.0–6.0)	n.s.	5.5 (3.0–6.7)	3.0 (3.0–5.5)	n.s.	4.0 (3.0–5.5)	4.0 (3.0–5.5)	n.s.	3–4 servings/week
Bread/paste/cereals (servings/day)	2.00 * (1.4–2.8)	2.7 (1.7–3.3)	0.005	2.1 (1.4–3.6)	2.9 (2.4–3.3)	n.s.	1.9 * (1.1–2.7)	2.3 (1.6–3.3)	0.022	4–6 servings/day
Pastries/desserts (servings/week)	4.2 (0.4–7.7)	4.0 (0.0–8.0)	n.s.	6.5 (2.2–8.5)	4.5 (1.5–8.0)	n.s.	4.0 (0.0–7.0)	3.0 (0.0–8.5)	n.s.	Optional, occasional, and moderate consumption
Nuts (servings/week)	1.5 (0.0–5.5)	0.0 (0.0–3.0)	n.s.	1.5 (0.0–6.0)	0.0(0.0–1.7)	n.s.	1.5 (0.0–5.5)	0.0 (0.0–4.0)	n.s.	3–7 servings/week
Beer/wine (servings/week)	1.5 (0.0–4.0)	1.5 (0.0–5.1)	n.s.	1.5 (0.2–5.5)	1.5 (0.0–5.5)	n.s.	1.5 (0.0–4.0)	1.5 (0.0–3.0)	n.s.	Optional, occasional, and moderate consumption

Results are expressed as median and range (P25-P75). * Significant differences (*p* ≤ 0.05) between celiac and non-celiac. ^1^ SENC: Spanish Society for Community Nutrition.

**Table 3 nutrients-13-01626-t003:** Energy, macronutrient distribution, expressed as contribution to total energy intake, fiber, and fatty acids in celiac and non-celiac Spanish men and women.

	Total Sample			Men			Women		
	Celiac (*n* = 64)	Non-Celiac (*n* = 74)	*p*	Celiac (*n* = 21)	Non-Celiac (*n* = 25)	*p*	Celiac (*n* = 43)	Non-Celiac (*n* = 49)	*p*
Energy (kcal/d)	1856.5 (1629.8–2134.5)	1799.0 (1570.0–2103.5)	n.s.	2082.0 (1773.5–2365.0)	1932.0 (1739.0–2154.0)	n.s.	1838.0 (1609.0–2031.0)	1691.0 (1433.0–2007.0)	n.s.
Energy (% of RI)	78.9 (67.5–87.5)	71.3 (62.2–86.1)	n.s.	75.3 (59.1–81.7)	64.7 (60.4–76.5)	n.s.	81.1 (73.2–89.7)	73.5 (64.5–88.4)	n.s.
Proteins(g/d)	80.8 (68.0–94.4)	79.4 (69.3–89.1)	n.s.	95.0 (80.1–114.0)	87.9 (79.5–96.0)	n.s.	78.3 (61.3–86.4)	73.1 (66.1–86.3)	n.s.
Proteins (% of RI)	182.2 (160.1–210.6)	174.9 (150.2–195.7)	n.s.	175.9 (148.3–211.1)	162.8 (147.1–177.7)	n.s.	187.1 (161.2–210.7)	178.3 (161.2–210.3)	n.s.
Proteins (% of TE)	16.7 (14.5–20.5)	17.5 (16.0–19.3)	n.s.	17.0 (14.5–21.4)	17.3 (16.2–19.20)	n.s.	16.7 (14.5–19.9)	17.6 (15.7–19.4)	n.s.
Total carbohydrates (% of TE)	38.9 (33.0–43.4)	38.2 (34.3–43.1)	n.s.	39.7 (33.4–44.5)	37.4 (33.9–41.25)	n.s.	38.8 (32.7–43.1)	39.0 (34.5–43.3)	n.s.
Simple sugars (% of TE)	17.1 (13.8–21.6)	16.4 (13.1–20.1)	n.s.	16.9 (12.6–22.0)	14.1 (11.9–18.0)	n.s.	18.1 (14.1–21.6)	17.3 (13.9–20.8)	n.s.
Fiber (g/day)	22.4 (15.6–26.9)	19.9 (15.7–26.3)	n.s.	25.7 (16.2–30.9)	19.4 (16.6–26.4)	n.s.	21.1 (14.9–25.5)	20.3 (14.7–26.3)	n.s.
Total lipids (% of TE)	39.3 (34.8–4.4)	40.3 (36.0–44.4)	n.s.	35.3 (34.1–46.1)	40.6 (38.0–44.3)	n.s.	40.1 (35.5–45.3)	39.8 (35.4–44.5)	n.s.
SFA (% of TE)	12.2 (10.5–14.2)	12.3 (9.9–14.0)	n.s.	11.2* (10.0–14.1)	13.3 (11.6–15.1)	0.046	12.6 (10.0–14.8)	11.4 (9.6–13.6)	n.s.
MUFA (% of TE)	15.6 (12.6–19-6)	17.4 (14.6–19.6)	n.s.	15.7 (12.8–19.2)	17.6 (14.9–19.2)	n.s.	15.5 (12.6–19.7)	16.6 (14.4–20.2)	n.s.
PUFA (% of TE)	4.9 * (3.7–6.4)	5.4 (4.4–6.7)	0.032	4.6 (3.7–6.7)	5.8 (5.2–6.8)	n.s.	4.9 (3.7–6.3)	5.1 (4.4–6.7)	n.s.
Cholesterol (mg/day)	316.0 (231.7–438.0)	312.5 (222.5–414.2)	n.s.	355.0 (251.0–509.5)	364.0 (279.5–440.5)	n.s.	308.0 (226.0–386.0)	292.0 (189.0–390.5)	n.s.
Trans fatty acids (mg/day)	87 (0.0–300)	140 (46–250)	n.s.	90 (46–355)	220 (120–290)	n.s.	84 (0.0–270)	92 (0.0–180)	n.s.
ω6 fatty acids (g/day)	2.0 (1.5–3.0)	2.3 (1.7–4.0)	n.s.	2.3 (1.7–4.5)	3.3 (2.0–5.3)	n.s.	1.9 (1.4–2.5)	1.9 (1.3–3.3)	n.s.
ω3 fatty acids (mg/day)	180 (102–240)	195 (140–292)	n.s.	210 (125–365)	250 (180–350)	n.s.	160 (0.100–0.210)	180 (130–265)	n.s.
EPA (mg/day)	103 (6–367)	145 (38–312)	n.s.	110 (7–435)	140 (39–295)	n.s.	200 (38–370)	200 (38–335)	n.s.
DHA (mg/day)	220 (88–712)	320 (92–600)	n.s.	240 (98–900)	260 (86–580)	n.s.	360 (89–640)	360 (89–625)	n.s.

Results are expressed as median and range (P25-P75). (% of TE): percentage contribution to total energy intake. * Significant differences (*p* ≤ 0.05) between celiac and non-celiac. SFA: Saturated Fatty Acids, MUFA: Monounsaturated fatty acids, PUFA: Polyunsaturated fatty acids, EPA: Eicosapentaenoic acid, DHA: Docosahexaenoic acid.

**Table 4 nutrients-13-01626-t004:** Percentage contribution of mineral intake to Recommended Intake in celiac and non-celiac Spanish men and women.

	Total Sample			Men			Women		
	Celiac (*n* = 64)	Non-Celiac (*n* = 74)	*p*	Celiac (*n* = 21)	Non-Celiac (*n* = 25)	*p*	Celiac (*n* = 43)	Non-Celiac (*n* = 49)	*p*
Calcium (% of RI)	71.0 (52.9–93.0)	72.7 (60.0–86.4)	n.s.	75.7 (55.1–107.9)	79.6 (63.7–90.5)	n.s.	68.9 (52.7–92.1)	69.6 (58.5–85.5)	n.s.
Phosphorus (% of RI)	164.8 * (136.1–194.6)	181.1 (156.2–203.4)	0.043	191.6 (165.4–232.6)	195.6 (181.8–214.5)	n.s.	148.7 * (130.9–177.1)	167.4 (141.1–197.8)	0.030
Iron (% of RI)	78.0 (53.3–130.5)	79.4 (67.8–132.0)	n.s.	133.0 (110.0–179.0)	148.0 (129.5–165.0)	n.s.	58.9 (49.4–78.3)	72.2 (61.9–80.8)	n.s.
Zinc (% of RI)	52.0 (46.2–66.5)	57.0 (47.1–65.5)	n.s.	67.3 (52.3–83.3)	60.7 (51.6–75.3)	n.s.	48.0 (42.7–56.7)	54.7 (46.0–63.6)	n.s.
Iodine (% of RI)	68.9 (51.7–89.5)	59.9 (49.7–83.8)	n.s.	64.4 (43.0–79.3)	56.5 (48.1–69.9)	n.s.	70.5 (55.9–90.9)	61.7 (51.7–85.5)	n.s.

Results are expressed as median and range (P25–P75). % of RI: Percentage contribution to recommended intake, calculated as actual intake/recommended intake × 100. * Significant differences (*p* ≤ 0.05) between celiac and non-celiac.

**Table 5 nutrients-13-01626-t005:** Percentage contribution of vitamin intake to Recommended Intake in celiac and non-celiac Spanish men and women.

	Total Sample			Men			Women		
	Celiac (*n* = 64)	Non-Celiac (*n* = 74)	*p*	Celiac (*n* = 21)	Non-Celiac (*n* = 25)	*p*	Celiac (*n* = 43)	Non-Celiac (*n* = 49)	*p*
Thiamine (% of RI)	114.6 (98.1–140.6)	126.1 (100.8–146.6)	n.s.	109.1 (89.55–148.10)	127.3 (100.0–145.5)	n.s.	120.0 (98.9–137.5)	122.2 (102.2–158.3)	n.s.
Riboflavin (% of RI)	100.0 (91.8–128.6)	107.1 (85.7–128.6)	n.s.	100.0 (76.50–126.05)	100.0 (76.5–117.1)	n.s.	107.1 (92.3–128.6)	107.7 (92.3–135.7)	n.s.
Pyridoxine (% of RI)	125.0 (100–187.0)	116.7 (100.0–131.3)	n.s.	138.9* (119.45–161.15)	116.7 (97.2–127.8)	0.015	118.8 (93.8–143.8)	112.5 (100.0–134.4)	n.s.
Vitamin B12 (% of RI)	242.5 (181.3–455.0)	237.5 (153.7–355.0)	n.s.	300.0 (195.0–530.0)	275.0 (187.5–480.0)	n.s.	230.0 (160.0–405.0)	215.0 (142.5–282.5)	n.s.
Niacin (% of RI)	103.6 (84.8–125.5)	98.9 (86.5–128.2)	n.s.	111.6 (85.5–125.8)	93.7 (78.7–105.6)	n.s.	102.9 (84.7–125.0)	101.3 (90.0–134.6)	n.s.
Folic Acid (% of RI)	64.7 (53.0–83.5)	61.25 (48.7–72.2)	n.s.	71.50 * (62.1–88.3)	59.5 (48.5–72.1)	0.032	61.3 (49.5–78.8)	62.8 (48.4–72.6)	n.s.
Vitamin C (% of RI)	243.30 * (180.0–304.2)	205.8 (134.3–254.5)	0.029	293.3 * (178.3–332.5)	196.7 (110.2–230.8)	0.012	226.7 (180.0–290.0)	211.7 (142.4–285.0)	n.s.
Vitamin A (% of RI)	122.05 * (89.2–152.9)	91.6 (69.8–117.4)	0.00	120.0 * (85.8–167.1)	79.0 (63.7–98.2)	0.001	123.0 * (92.6–148.3)	101.9 (73.4–137.9)	0.028
Vitamin D (% of RI)	21.35 (11.3–47.0)	22.7 (9.1–36.3)	n.s.	27.3 (12.0–56.3)	22.0 (9.0–35.6)	n.s.	20.0 (9.30–46.0)	22.7 (9.0–37.3)	n.s.
Vitamin E (% of RI)	69.60 (57.7–83.1)	62.1 (49.2–85.6)	n.s.	81.7 (66.7–88.7)	67.5 (54.6–96.6)	n.s.	65.8 (56.7–77.5)	60.0 (46.2–75.0)	n.s.
Vitamin K (% of RI)	138.9 * (103.4–219.8)	109.0 (77.5–165.2)	0.006	149.2 * (118.7–185.0)	100.00 (60.0–155.8)	0.024	135.6 (100.6–242.2)	115.6 (84.2–172.7)	n.s.

Results are expressed as median and range (P25–P75). % of RI: Percentage contribution to recommended intake, calculated as actual intake/recommended intake × 100 * Significant differences (*p* ≤ 0.05) between celiac and non-celiac groups.

**Table 6 nutrients-13-01626-t006:** Anthropometric measurements in celiac and non-celiac Spanish men and women.

	Total Sample			Men			Women		
	Celiac (*n* = 64)	Non-Celiac (*n* = 74)	*p*	Celiac (*n* = 21)	Non-Celiac (*n* = 25)	*p*	Celiac (*n* = 43)	Non-Celiac (*n* = 49)	*p*
Weight (kg)	64.0 (55.4–80.6)	64.6 (54.2–73.8)	n.s.	72.3 * (65.4–76.4)	81.4 (72.9–89.3)	0.003	57.4 (51.7–66.5)	56.0 (51.2–64.6)	n.s.
Height (cm)	167.9 (162.0–183.1)	166.9 (160.4–179.9)	n.s.	179.6 (171.7–184.2)	177.5 (174.2–180.2)	n.s.	163.8 (158.0–168.6)	163.1 (158.1–166.9)	n.s.
Body Fat (%)	30.6 (22.1–38.7)	28.7 (23.2–38.7)	n.s.	21.7 * (17.5–26.3)	26.0 (21.6–29.1)	0.019	33.4 * (29.8–36.5)	29.6 (25.0–35.0)	0.027
BMI (kg/m^2^)	21.8 (20.2–27.5)	22.7 (20.5–28.2)	n.s.	22.6 * (21.2–24.1)	24.9 (23.3–27.9)	0.000	21.3 (20.0–23.9)	21.7 (18.9–23.2)	n.s.

Results are expressed as median and range (P25–P75). BMI: Body Mass Index. * Significant differences (*p* ≤ 0.05) between celiac and non-celiac.

**Table 7 nutrients-13-01626-t007:** Bone mineral density in celiac and non-celiac Spanish men and women.

	Total Sample			Men			Women		
	Celiac (n = 64)	Non-Celiac (*n* = 74)	*p*	Celiac (*n* = 21)	Non-Celiac (*n* = 25)	*p*	Celiac (*n* = 43)	Non-Celiac (*n* = 49)	*p*
BMD (g/cm^2^)	0.520 (0.440–0.610)	0.560 (0.460–0.650)	n.s.	0.540 (0.450–0.650)	0.630 (0.530–0.700)	n.s.	0.510 (0.440–0.600)	0.530 (0.440–0.620)	n.s.

Results are expressed as median and range (P25–P75). BMD: Bone Mineral Density.

**Table 8 nutrients-13-01626-t008:** Hemogram in celiac and non-celiac Spanish men and women.

	Total Sample			Men			Women			
	Celiac (*n* = 64)	Non-Celiac (*n* = 74)	*p*	Celiac (*n* = 21)	Non-Celiac (*n* = 25)	*p*	Celiac (*n* = 43)	Non-Celiac (*n* = 49)	*p*	Reference Value
Red blood cells/μL (×10^6^)	4.64 (4.4–4.8)	4.73 (4.4–5.1)	n.s.	4.98* (4.7–5.0)	5.17 (4.9–5.4)	0.022	4.5 (4.2–4.7)	4.6 (4.3–4.8)	n.s.	M:4.6–6.20 W:4.20–5.40/μL (×10^6^)
Hemoglobin (g/dL)	14.30 (13.6–15.1)	14.20 (13.5–15.2)	n.s.	15.20 (14.9–15.8)	15.50 (14.8–16.3)	n.s.	13.7 (13.3–14.3)	13.8 (13.0–14.3)	n.s.	M:13.5–18 W:12–16 g/dL
Hematocrit (%)	42.5 (40.0–44.2)	42.6 (39.5–45.0)	n.s.	45.0 (43.9–46.2)	45.7 (43.9–48.0)	n.s.	40.9 (39.6–42.9)	41.1 (38.9–42.9)	n.s.	M:42–52% W:37–47%
MCV (µ^3^)	91.1 (88.5–94.5)	90.4 (86.7–93.4)	n.s.	91.7 * (89.8–95.6)	89.8 (86.1–92.4)	0.031	90.4 (88.0–94.0)	90.6 (86.8–94.2)	n.s.	80–96 (µ^3^)
MCH (pg)	30.7 (30.1–31.9)	30.4 (29.1–31.6)	n.s.	31.2 * (30.3–32.7)	30.3 (29.3–31.5)	0.018	30.7 (30.0–31.7)	30.5 (28.9–31.7)	n.s.	27–33 pg
MCHC (%)	33.9 (33.2–34.3)	33.6 (33.0–34.2)	n.s.	34.1 (33.3–34.5)	33.8 (33.4–34.5)	n.s.	33.7 (33.0–34.3)	33.3 (32.9–34.0)	n.s.	33–37%
RDW (%)	11.7 (11.5–12.2)	11.9 (11.6–12.5)	n.s.	11.7 (11.4–12.1)	11.9 (11.5–12.2)	n.s.	11.9 (11.5–12.2)	12.1 (11.6–12.7)	n.s.	11–18%
Platelets /μL (×10^3^)	219.0 (181.7–247.0)	214.5 (190.7–256.2)	n.s.	195.0 (182.0–235.5)	206.0 (185.5–226.5)	n.s.	230.0 (181.0–318.6)	229.0 (199.5–299.0)	n.s.	130–450/μL (×10^3^)
MPV (µ^3^)	8.5 * (8.0–9.2)	8.8 (8.1–9.3)	0.126	8.6 (8.1–9.1)	9.0 (8.2–9.2)	n.s.	8.4 (7.9–9.2)	8.8 (8.0–9.3)	n.s.	7–13 (µ^3^)
Leukocytes /μL (×10^3^)	5.5 * (4.6–6.6)	6.2 (5.0–7.0)	0.014	5.5 * (4.6–6.2)	6.8 (5.4–7.8)	0.003	5.6 (4.6–6.7)	5.8 (5.0–6.8)	n.s.	4.00–11.00/μL (×10^3^)
Lymphocytes /μL	1837.00 (1567.0–2326.0)	2017.5 (1728.7–2393.0)	n.s.	1740.0 * (1571.0–2246.0)	2100.0 (1732.5–2522.0)	0.050	1941.0 (1561.0–2385.0)	1994.0 (1705.0–2361.5)	n.s.	1000–4500/μL
Monocytes /μL	426.0 (362.5–487.2)	456.5 (376.2–534.5)	n.s.	428.0 * (353.0–478.0)	473.0 (411.0–578.0)	0.050	424.0 (362.0–528.0)	446.0 (357.0–525.0)	n.s.	1800–7500/μL
Neutrophils /μL	2926.5 * (2258.5–3567.5)	3361.5 (2645.7–4153.0)	0.024	3038.0 * (2280.0–3458.5)	3452.0 (2914.5–4579.5)	0.028	2854.0 (2230.0–3684.0)	3150.0 (2537.5–3962.5)	n.s.	<800/μL
Eosinophils /μL	148.5 (82.7–225.7)	163.5 (100.2–241.0)	n.s.	167.0.0 (84.5–225.0)	195.0 (107.5–245.0)	n.s.	148.0 (77.0–239.0)	150.0 (99.5–241.5)	n.s.	<800/μL
Basophils /μL	43.0 (23.5–61.5)	51.0 (38.0–63.2)	n.s.	43.0 (22.5–58.5)	51.0 (37.0–74.5)	n.s.	40.0 (25.0–63.0)	52.0 (39.0–62.5)	n.s.	<200/μL

Results are expressed as median and range (P25-P75). * Significant differences (*p* ≤ 0.05) between celiac and non-celiac. MCV: Mean Corpuscular Volume; MCH: Mean Corpuscular Hemoglobin; CMHC: Mean Corpuscular Hemoglobin Concentration; RDW: Red Cell Distribution Width. VPM: Medium Platelet Volume.

**Table 9 nutrients-13-01626-t009:** Plasma levels of biochemical parameters in celiac and non-celiac Spanish men and women.

	Total Sample			Men			Women			
	Celiac (*n* = 64)	Non-Celiac (*n* = 74)	*p*	Celiac (*n* = 21)	Non-Celiac (*n* = 25)	*p*	Celiac (*n* = 43)	Non-Celiac (*n* = 49)	*p*	Reference Value
Basal Glucose (mg/dL)	83.0 (77.0–88.0)	81.5 (76.7–87.2)	n.s.	86.0 (78.0–90.0)	84.0 (79.0–91.0)	n.s.	82.0 (77.0–88.0)	80.0 (75.0–86.0)	n.s.	60–110 mg/dL
Albumin (g/dL)	4.5 (4.3–4.7)	4.5 (4.3–4.6)	n.s.	4.7* (4.5–4.9)	4.5 (4.4–4.6)	0.039	4.5 (4.3–4.6)	4.5 (4.3–4.6)	n.s.	3.5–5.2 g/dL
Iron (μg/dL)	100.5 (81.2–121.7)	107.5 (84.7–132.0)	n.s.	106.0 (90.5–133.0)	108.0 (91.0–128.5)	n.s.	97.0 (76.0–113.0)	104.0 (81.0–133.5)	n.s.	37–160 (μg/dL)
Folate (ng/mL)	8.0 (4.3–10.3)	6.3 (4.60–8.62)	n.s.	6.7 (4.0–9.9)	5.5 (4.4–7.3)	n.s.	8.1 (5.7–10.7)	7.0 (4.8–9.2)	n.s.	3–17 ng/mL
Homocysteine (µmol/L)	10.0 (8.4–11.1)	9.6 (8.1–12.1)	n.s.	11.5 (9.7–13.0)	11.7 (9.7–13.4)	n.s.	9.7 (8.4–10.6)	8.8 (7.7–10.4)	n.s.	<15.4 µmol/L
Calcium (mg/dL)	9.4 (9.1–9.6)	9.3 (9.0–9.6)	n.s.	9.4 (9.1–9.5)	9.4 (9.1–9.6)	n.s.	9.4 (9.0–9.6)	9.3 (8.9–9.6)	n.s.	8.2–10.6 mg/dL
Phosphorus (mg/dL)	3.7 (3.4–3.9)	3.6 (3.3–4.0)	n.s.	3.5 (3.0–3.7)	3.4 (3.2–3.6)	n.s.	3.8 (3.7–4.0)	3.8 (3.4–4.1)	n.s.	2.5–5 mg/dL
Cholesterol (mg/dL)	183.5 (148.2–205.5)	177.5 (155.0–206.0)	n.s.	191.0 (145.0–220.0)	186.0 (164.0–210.5)	n.s.	180.0 (152.0–201.0)	174.0 (149.0–204.5)	n.s.	<200 mg/dL
Triglycerides (mg/dL)	59.0 (47.25–74.0)	70.5 (50.5–97.5)	0.044	61.0 (46.5–94.5)	72.0 (60.0–113.5)	*n.s.*	59.0 (48.0–69.0)	64.0 (48.0–95.0)	n.s.	<200 mg/dL
HDL (mg/dL)	63.0 (51.7–73.0)	65.5 (54.2–74.2)	n.s.	56.0 (47.5–68.0)	52.0 (44.0–67.5)	n.s.	66.0 (56.0–74.0)	68.0 (61.0–77.0)	n.s.	>40 mg/dL
LDL (mg/dL)	100.0 (83.2–123.0)	96.0 (76.7–124.0)	n.s.	104.0 (76.0–135.0)	111.0 (96.0–133.5)	n.s.	99.0 (84.0–112.0)	87.0 (69.5–115.0)	n.s.	<130 mg/dL: Primary prevention <100 mg/dL Secondary prevention
Cholesterol/HDL	2.8 (2.4–3.2)	2.7 (2.2–3.4)	n.s.	2.8 (2.5–3.4)	3.4 (2.8–4.2)	n.s.	2.8 (2.4–3.1)	2.5 (2.1–2.9)	n.s.	<4.5
LDL/HDL	1.6 (1.3–1.9)	1.4 (0.97–2.15)	n.s.	1.6 (1.4–2.1)	2.1 (1.6–2.8)	n.s.	1.5 * (1.3–1.9)	1.3 (0.9–1.7)	0.032	M: <3.55 W: <3.22
Vitamin D (ng/mL)	34.7 (24.5–59.9)	33.7 (22.2–66.8)	n.s.	43.7 (26.8–55.3)	31.6 (19.8–63.2)	n.s.	33.8 (22.1–60.6)	38.0 (22.3–68.8)	n.s.	<10 ng/mL: Moderate deficit; 10–30 ng/mL: Severe deficit; 30–96 ng/mL Recommended values; >96 ng/mL excess
Parathormone (pg/mL)	32.85 (24.0–51.3)	34.20 (18.9–48.8)	n.s.	31.30 (24.9–54.9)	36.7 (18.8–46.6)	n.s.	32.9 (23.4–29.9)	34.2 (18.8–51.5)	n.s.	14.5–87.1 pg/mL

Results are expressed as median and range (P25-P75). * Significant differences (*p* ≤ 0.05) between celiac and non-celiac.

## Data Availability

The data presented in this study are available on request from the corresponding author.

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
