# Peer review of "Nutritional Status in Spanish Adults with Celiac Disease Following a Long-Term Gluten-Free Diet Is Similar to Non-Celiac"

_nutrients, 2021, doi:10.3390/nu13051626_

Round 1
Reviewer 1 Report
This paper is a worthwhile contribution to the field, in which the nutritional status of Spanish adults with Celiac disease following 1-year of gluten free diet is compared to non-celiac adults. Overall, the paper is well presented with a good summary of data. However, there are several aspects of this paper that warrant attention from the authors for this paper to reach its full potential.
Abstract
Line 11 Please confirm the study is age-matched? Method states gender-matched subjects only.
Line 21-23 Should be the last sentence of the paragraph.
Introduction
Line 53 I don’t think conditioned is the right word. Please revise.
Line 83 Please revise to improve readability.
Line 103-107 Revise aim. The study is a cross-sectional section with no baseline measurements. I don’t think long term effects of GFD can be evaluated with the current study design.
When stating contrasting study results. Please include differences/similarities in the study design which may have contributed to the mixed results.
Materials and Methods
Was there any power analysis done?
Line 194 Change red blood cells to red blood cell count; platelets to platelet count; leuckocytes, lymphocytes, monocytes, neutrophils, eosinophils, basophils to leukocyte differential count (leukocytes, lymphocytes, monocytes, neutrophils, eosinophils, basophils)
Line 221 Age groups not presented in the results (except for Figure 1)?
Results
Line 238 Does this mean that others did not follow a strict gluten free diet? Should these participants be excluded or an inclusion criteria added in terms of how much the participants followed the diet?
Line 240 Data showed fruit intake to have no difference between groups (higher in celiac men vs non-celiac men, though not significant)
Line 242-243 Vegetable drinks, pickles not in the table (Table 2)
Table 2-9 Please ensure P column entry aligns with the rest of the table or this column can be deleted with the P values provided in the text. I would also suggest adding one column (to appropriate tables) which provides recommended dietary intakes for comparison (similar to what was provided in Table 2 – SENC recommendations).
Line 288 PUFA significantly lower in celiac group compared to non-celiac group – please include in the text.
Line 330 thinness is not the appropriate term. Please use underweight.
Figure 1. Is the data presented for the group overall since the groups were categorized by age? Instead of categorizing body weight to age groups, it would be better to perform ANCOVA or ANCOVRES whichever is applicable adjusting for confounding variables (age, body fat composition).
Line 360 What do you mean by prevalence was similar in both groups? Please clarify.
Line 391 What do you mean by not significative? Reference for the prevalence values?
Line 404-405 Please provide classification/cut-off used in terms of METs level.
Discussion
Line 44 Describe the studies, what type of study design? Sample sizes? Compare with your study design.
Line 51 Change nonetheless to interestingly
Line 72 Describe the studies, what type of study design? Sample sizes? Compare with your study design.
Line 86 Change within the ranges of normality to within the reference ranges.
General comment: When comparing previous studies with your current study, please include differences/similarities in the study design which may have contributed to the mixed results or possible explanation for the differing results.
Strengths and Limitations
Low sample size should be added as a limitation of the study.
Author Response
Reviewer 1
We wish to thank the reviewer for his/her comments that for sure will increase the quality of our paper. We hope you will find the new version more satisfactory. Changes are highlighted using track changes in the document.
Abstract
- Line 11 Please confirm the study is age-matched? Method states gender-matched subjects only.
- Yes, the study is age-matched. The issue has been clarified in the Methods section. Line 21-23 Should be the last sentence of the paragraph.
According to the reviewer suggestion, we have moved line 21-23 to the end of the paragraph.
Introduction
- Line 53 I don’t think conditioned is the right word. Please revise.
According to the reviewer suggestion, we changed this word to “explained”
- Line 83 Please revise to improve readability.
We have rewritten the paragraph in order to improve the understanding of the text.
- Line 103-107 Revise aim. The study is a cross-sectional section with no baseline measurements. I don’t think long term effects of GFD can be evaluated with the current study design.
First, we wish to thank the reviewer for this comment. The people who participated in the study are long-term gluten-free dieters. The study assesses the nutritional status of this population. It is true that the design of the study is not longitudinal, so we have rephrased the aim. There is sufficient basis in the scientific literature to state that the nutritional status of celiac patients at the time of diagnosis of the disease is extremely unbalanced. After one year following a GFD, disease symptoms disappear, and a period over one year is reasonably sufficient to consider that the adherence to GFD is long-term, to bring about changes in the dietary patterns, nutritional and health status of this population.
- When stating contrasting study results. Please include differences/similarities in the study design which may have contributed to the mixed results.
We added a sentence in line 73 that clarifies this question.
Materials and Methods
- Was there any power analysis done?
We used power analysis to calculate the minimum sample size to be able to observe any differences between celiac and non-celiac. Taking into account previous studies in Spain and considering a confidence interval of 95%, an error a of 5%, an error b of 20%, a power of 80% and a case: control ratio of 1:1; using the EpiInfo v.7 software, a total sample of 110 adults was calculated. In order to account for losses, we added a 20% so an initial sample of 75 cases and 75 controls aged 19-59 years was proposed. Finally, the sample was slightly reduced as 8% of individuals with celiac disease reported digestive discomfort in the days prior to the study. Power analysis has been added to the manuscript.
- Line 194 Change red blood cells to red blood cell count; platelets to platelet count; leuckocytes, lymphocytes, monocytes, neutrophils, eosinophils, basophils to leukocyte differential count (leukocytes, lymphocytes, monocytes, neutrophils, eosinophils, basophils)
According to the reviewer suggestion, we made the changes.
- Line 221 Age groups not presented in the results (except for Figure 1)?
The results are not presented by age groups except when individuals are classified according to body fat categories (this issue is explained in the text just after figure 1). All parameters not presented by age groups are those that have reference values that are the same regardless of age (i.e., BMI, dietary goals, recommended food rations, etc.). In the case of the adequacy of nutrient intakes to recommended intakes, the percentage of compliance is calculated using the recommended intake for gender an age in each case. Therefore, we only refer to the categorization by age group (young adults and older adults) in the case of body fat, since reference values are different according to age.
Results
- Line 238 Does this mean that others did not follow a strict gluten free diet? Should these participants be excluded or an inclusion criteria added in terms of how much the participants followed the diet?
The criteria for inclusion in the study were “adherence to a gluten-free diet for more than one year and not objectively presenting gastrointestinal symptoms or any other symptom of celiac disease in the active phase”. IgA transglutaminase antibodies testing was used to exclude participants with undiagnosed celiac disease or a low adherence to a gluten free diet. A questionnaire on the personal perception of the gluten-free diet was added, which included some questions that assessed the diet qualitatively. The only way to confirm adherence to the gluten-free diet is through the measurement of IgA class tissue transglutaminase antibodies, as well as through the declaration of the participants.
Indeed, the reality is that not all celiac patients follow a strict gluten-free diet (6.2% of them answered NO to the question in the questionnaire); however, we do not consider that these patients should be excluded from the study because the testing for IgA transglutaminase antibodies was negative.
- Line 240 Data showed fruit intake to have no difference between groups (higher in celiac men vs non-celiac men, though not significant)
According to the reviewer suggestion we removed fruits in the sentence.
- Line 242-243 Vegetable drinks, pickles not in the table (Table 2)
We appreciate this comment. Both vegetable drinks and pickles were relatively infrequently consumed in the two groups, so that the median calculation was very low in some cases (0.0 servings per week). However, when calculating the frequency of individuals who consumed these foods at least once a week, it could be observed that this intake was higher in the coeliac population (100% vs. 77% for vegetable drinks and 69% vs. 41% for pickles) and this difference was statistically significant. We believe that the latter way of expressing the data is more clarifying and understandable. This is the reason why this data ar not represented in the table.
- Table 2-9 Please ensure P column entry aligns with the rest of the table or this column can be deleted with the P values provided in the text. I would also suggest adding one column (to appropriate tables) which provides recommended dietary intakes for comparison (similar to what was provided in Table 2 – SENC recommendations).
According to the reviewer´s suggestion we modified the P column in tables 2-9. In the tables showing the adequacy of nutrients to the recommended intakes and variables that form part of the nutritional objectives for the Spanish population, we have not added a column of recommendations because they are cited throughout the text of the results. In some cases, recommendations are different depending on gender an age and, thus, making it very difficult to include in the table.
- Line 288 PUFA significantly lower in celiac group compared to non-celiac group – please include in the text.
Following the reviewer’s suggestion, we included this sentence in the text.
- Line 330 thinness is not the appropriate term. Please use underweight.
As kindly suggested, the term thinness has been replaced by underweight.
- Figure 1. Is the data presented for the group overall since the groups were categorized by age? Instead of categorizing body weight to age groups, it would be better to perform ANCOVA or ANCOVRES whichever is applicable adjusting for confounding variables (age, body fat composition).
Thank you for your proposal, which we find very interesting. However, we felt that it was not necessary to carry out the statistical tests, since, by categorizing by age group in the assessment of body fat, we were already eliminating the confounding factor.
- Line 360 What do you mean by prevalence was similar in both groups? Please clarify.
When we compared the prevalence of individuals with osteopenia or osteoporosis and those with normal bone mass, according to the T-score, between men with coeliac disease and men in the control group, the Pearson's chi-square test indicated that they were not significantly different, indicating that the distribution of the sample of celiac and controls was similar between the three groups we have established (osteoporosis, osteopenia, normality).
- Line 391 What do you mean by not significative? Reference for the prevalence values?
Thank you for your comment. In order to clarify the text, we rephrased the sentence. As explained in Subjects and Methods, biochemical analysis of the samples was performed at the Megalab laboratories, who provided the reference values according to the technology used.
- Line 404-405 Please provide classification/cut-off used in terms of METs level.
As described in the methodology for analyzing the questionnaire, depending on the intensity of the activity performed (walking, moderate activity or vigorous activity), the time spent is multiplied by a factor (3.3; 4 or 8, respectively), and by the number of days on which the activity was performed, so that we obtained a final result in METS (minute/week), which corresponded to the metabolic equivalents used and are representative of the intensity and duration of the physical activity performed. This calculation allows us to categorize the physical activity performed as insufficient, moderate or high, according to the following ranges:
- High or vigorous physical activity: this category describes the highest levels of activity during the week in question. The IPAQ Research Committee, in the absence of common criteria for defining vigorous activity, establishes the criterion for high physical activity by considering:
- Vigorous physical activity at least 3 three days per week achieving a total of at least 1,500 METs.
- Seven days of any combination of walking, moderate physical activity and/or vigorous physical activity, achieving a total of 3,000 METs.
- Moderate physical activity:
- Three or more days of vigorous physical activity for at least 20 twenty minutes per day.
- Five or more days of moderate physical activity and/or walking for at least 30 thirty minutes per day.
- Five or more days of any combination of walking, moderate or vigorous physical activity achieving at least a total of 600 METs.
- Insufficient or low physical activity: this category is defined as not meeting the criteria of the previous categories.
Discussion
- Line 44 Describe the studies, what type of study design? Sample sizes? Compare with your study design.
More information on the studies has been introduced in the text.
- Line 51 Change nonetheless to interestingly
As kindly suggested, this correction has been made.
- Line 72 Describe the studies, what type of study design? Sample sizes? Compare with your study design.
More information on the studies has been introduced in the text.
- Line 86 Change within the ranges of normality to within the reference ranges.
As kindly suggested, this correction has been made.
- General comment: When comparing previous studies with your current study, please include differences/similarities in the study design which may have contributed to the mixed results or possible explanation for the differing results.
As you have suggested, we have made comparisons more specific where possible.
Strengths and Limitations
- Low sample size should be added as a limitation of the study.
According to the reviewer´s suggestion, in the new version of the manuscript we added this limitation.
Reviewer 2 Report
The Study analysed the complete evaluation of the nutritional status, through dietary and body composition analysis, and biochemical and physical activity measures, in Spanish adults diagnosed with celiac disease; data were compared with non-celiachealthy subjects, according to gender, todetect possible deficiencies and nutritional imbalances in celiac population.
Overall, the Study provided a meaningful contribution to the field of
research. The science underlying the main idea is strong. There are no
significant gaps in the cited literature. The results are original and allow
incremental advance over prior research results. The research design is
appropriate. The right kinds of participants were used. The sample size was
adequate. The correct statistics were used. The interpretation of the data
makes sense and logically supports the conclusions. The findings are
important and interesting to the readers. The methods are explained well
enough that the experiments can be replicated. The discussion section
intgrates the findings with relevant theory, rather than simply rehashing the introduction.
Author Response
Reviewer 2
We really appreciate the reviewer’s analysis of the manuscript. Thank you very much. We have made a final English editing to improve it.